# Pyrimidines-Based Heterocyclic Compounds: Synthesis, Cytoxicity Evaluation and Molecular Docking

**DOI:** 10.3390/molecules27154912

**Published:** 2022-08-01

**Authors:** Mohamed A. El-Atawy, Najla A. Alshaye, Nada Elrubi, Ezzat A. Hamed, Alaa Z. Omar

**Affiliations:** 1Chemistry Department, Faculty of Science, Taibah University, Yanbu 46423, Saudi Arabia; 2Chemistry Department, Faculty of Science, Alexandria University, P.O. Box 426 Ibrahemia, Alexandria 21321, Egypt; nadaelrubya.allah@gmail.com (N.E.); ezzat.awad@alexu.edu.eg (E.A.H.); 3Department of Chemistry, College of Science, Princess Nourah Bint Abdulrahman University, Riyadh 11671, Saudi Arabia; naalshaye@pnu.edu.sa

**Keywords:** pyrimidine, MTT, cytotoxicity, molecular docking, prostate cancer, PC3

## Abstract

A variety of structurally different pyrimidines were synthesized. Elemental analysis, FT-IR, ^1^H NMR, and ^13^C NMR spectroscopy were used to confirm the chemical structures of all prepared compounds. The synthesized pyrimidines were screened against the growth of five human cancer cell lines (prostate carcinoma PC3, liver carcinoma HepG-2, human colon cancer HCT-116, human breast cancer MCF-7, human lung cancer A-549), and normal human lung fibroblasts (MRC-5) using MTT assay. Most of the screened pyrimidines have anti-proliferative activity on the growth of the PC3 cell line. Compounds **3b** and **3d** were more potent than the reference vinblastine sulfate (~2 to 3 × fold) and they can be considered promising leads for treating prostate cancer disease. Moreover, the screened compounds **3b**, **3f**, **3g**, **3h**, and **5** were assessed according to the values of their selectivity index (SI) and were found to be more selective and safer than vinblastine sulfate. Furthermore, using in silico computational tools, the physicochemical properties of all pyrimidine ligands were assessed, and the synthesized compounds fall within the criteria of RO5, thus having the potential to be orally bioavailable.

## 1. Introduction

The pyrimidine skeleton is the parent substance of many vital compounds that occur in nature, particularly in nucleobases of nucleic acids such as cytosine, thymine, and uracil. Moreover, pyrimidine scaffolds are readily accessible through the Biginelli reaction as well as possess a wide diversity of pharmacological and therapeutic properties including antimicrobial [1], antibacterial [2], antifungal [3], herbicidal activity [4], anti-inflammatory [5], therapeutic potentiality [6], antitubercular [7], anticancer [8], anticonvulsant [9], antileishmanial [10], antihypertensive [11,12], cytotoxicity [13], and antitumor [14]. Tarceva^®^ (Erlotinib) is one of the most often used pyrimidine-based anticancer medicines, as seen in Figure 1. Which has been approved by the FDA for the medical treatment of pancreatic cancer and non-small cell lung cancer (NSCLC). Moreover, triazolopyrimidines derivative (Trapidil) behaves as a platelet-derived growth factor antagonist for inhibition of parathyroid bone disease [15] and as a phosphodiesterase inhibitor for the treatment of memory disorders [16].

Anticancer activity of pyrimidine derivatives has been reported using a variety of mechanisms, including antiproliferative activity [17], tubulin polymerization inhibition [18], topoisomerase inhibition [19], epidermal growth factor receptor (EGFR) inhibitors [20], phosphatidylinositol 3-kinases (PI3K) inhibitor [21], mTOR/ PI3Ka inhibitors [22], EGFR/ErbB-2 KINASE dual inhibitors [23], c-Met/VEGFR-2 dual inhibitors [24], aurora kinases inhibitors [25], etc. Furthermore, hydrazones, which are easily accessible from heterocyclic hydrazides and aromatic or heteroaromatic aldehydes, have also been explored as promising anticancer agents either chelated with metals [26] or in the free form [27].

Furthermore, Prostate cancer is a growing concern worldwide. It is ranked as the second most common type of cancer and the fifth leading cause of mortality in men [28]. Prostate cancer cells could spread to other parts of the body, predominantly the lymph nodes and bones [29]. Consequently, the research and development of new anticancer drugs remains a challenge in the field of medicinal chemistry.

The current research aims to develop new pyrimidine derivatives that are easily synthesized from readily accessible starting materials at mild conditions with good yields and evaluate their in vitro cytotoxicity against a number of human cancer cell lines. Part of the work was devoted to the prediction of physicochemical properties, drug-likeness, and oral bioavailability in silico. Moreover, molecular docking studies were used to evaluate the interaction of the synthesized pyrimidines with B-cell lymphoma 2 (Bcl-2) active sites. This would be economically more advantageous in terms of finding and discovering the most active drug. Finally, as new chemical entities, the MTT assay was used to assess the in vitro cytotoxicity of the synthesized pyrimidines against the growth of five human cancer cell lines (prostate carcinoma PC3, liver carcinoma HepG-2, human colon cancer HCT-116, human breast cancer MCF-7, human lung cancer A-549), and normal human lung fibroblasts (MRC-5).

## 2. Results and Discussion

### 2.1. Chemistry

The pyrimidine ligands were synthesized as outlined in Figure 1. Compound **1** has been prepared via the Biginelli cyclocondensation reaction of ethyl cyanoacetate, benzaldehyde, and thiourea in presence of ethanolic potassium carbonate followed by acidification by acetic acid.

The synthetic strategy (Figure 2) for pyrimidines was based on the presence of a 2-thioxopyrimidine-4-one moiety in compound **1** as analogs for the 2-thioxoimidazolidin-4-one moiety (Figure 3), which is found in several FDA-approved anticancer drugs [30]. As a result, compound **1** was chosen as a lead for new pyrimidines with probable biological activity or anticancer efficiency. A second aromatic ring was incorporated at position 2 via a hydrazine linker. Furthermore, the addition of a third aromatic ring as well as cyclization of the linker has been chosen to improve the hydrophobic character. Alternatively, polar interaction has been enhanced by designing compound **6** with only one aromatic ring and polar OH and NH groups. In general, the presence of the aromatic rings, as well as polar groups such as cyano, hydroxyl, or amino groups, fulfills the hydrophobic and polar interactions required for inhibitory activity.

The ^1^H NMR spectrum of **1** showed a broad exchangeable signal at δ 12.82 ppm that corresponds to NH protons, a doublet at δ 7.66, triplet at δ 7.63, and triplet δ 7.56 ppm that can be ascribed to ortho, para, and meta protons of the benzene ring, respectively. Moreover, APT ^13^C NMR exhibited signals at δ 176.30, 161.06, and 113.81 ppm respective to carbons of thiocarbonyl, carbonyl, and carbonitrile groups, respectively. The reaction of **1** with hydrazine hydrate afforded hydrazine **2**. Subsequent reaction of **2** with a variety of aromatic aldehydes or isatin yielded the corresponding hydrazones **3a**–**i** and **4** in good yields. Furthermore, cyclocondensation of **2** with either benzoin or chloroacetyl chloride afforded compounds **5** and **6**, respectively. The infrared absorption of **3a**–**i** showed bands in the region 3445–3207 cm^−1^ ascribed to the NH group, whereas the cyano group exhibited a medium absorption band at 2204–2237 cm^−1^. Moreover, the carbonyl of the lactam moiety showed strong intensity bands at 1695–1652 cm^−1^. Furthermore, compounds **3e** and **3f** exhibited additional absorption bands at 2957 and 2922 cm^−1^, respectively, which corresponds to stretching of the aliphatic C-H bond, whereas **3g** exhibited an additional broad absorption band at 3439 cm^−1^ corresponding to vibration of the hydroxyl group. The ^1^H NMR spectral data of **3a**–**i** were recorded in DMSO-*d6*. As a prototype, the ^1^H-NMR spectrum of compound **3e** revealed the presence of two singlets at δ 12.59 and 12.39 ppm due to the existence of pyrimidine NH and hydrazone NH, respectively. The singlet at δ 8.15 ppm corresponds to azomethine (CH=N) proton, a multiplet at 7.85–7.01 ppm was assigned to aromatic protons, and a singlet at 3.82 ppm for the methyl group. Moreover, the ^1^H-NMR spectrum of the **3g** revealed the existence of a broad singlet at 9.99 ppm for OH proton. The three allyl protons of compound **3h** appeared as a doublet of doublet and two other doublets at 6.98, 7.15, and 8.00, respectively. The IR spectra of **4** showed an absorption band at 3439 cm^−1^ corresponding to NH vibration, and a band at 2225 cm^−1^ for the cyano group. Furthermore, the carbonyl of the two lactam groups appeared at 1685, 1627 cm^−1^. The ^1^H NMR spectrum of compound **4** in DMSO-*d6* showed two exchangeable broad singlets at δ 12.78 and 11.31 ppm which were assigned to the three NH of pyrimidine, hydrazino, and isatin, respectively. The protons of the aromatic ring derived from isatin appeared as two doublets at δ 7.80 and 6.93 and two triplets at δ 7.38 and 7.12. The IR spectrum of **5** showed absorption bands at 3351, 2216, 1600, and 1588 cm^−1^ correspond to the stretching vibrations of NH, C≡N, C=O, and C=N groups, respectively. Moreover, the ^1^H NMR spectrum in DMSO-*d6* exhibited an exchangeable signal at δ 12.45 ppm assigned for the NH proton, singlet at δ 6.46 ppm corresponding to CH of the triazine ring. In addition, the IR spectrum of **6** showed a broad band at 3416 cm^−1^ assigned to the vibrations of the OH group, and a weak absorption band at 3300 cm^−1^ assigned to the vibration of the NH group. Bands at 2207, 1649, and 1595 cm^−1^ corresponded to the vibration of cyano, carbonyl, and imine groups. The ^1^H NMR spectrum of **6** in DMSO-*d6* showed two broad exchangeable signals at δ 11.00 and 7.78 ppm corresponding to NH and OH protons, respectively. The aromatic protons appeared as a doublet at δ 7.66 ppm and multiplet at δ 7.52.

### 2.2. Cytotoxicity and Anticancer Activity

Preliminary in vitro anticancer activity of two compounds (**2**, and **3a)** against the growth of five human cancer cell lines (prostate carcinoma PC3, liver carcinoma HepG-2, human colon cancer HCT-116, human breast cancer MCF-7, human lung cancer A-549) and normal human lung fibroblasts (MRC-5) were performed using the MTT assay as described in the experimental section. The inhibitory potency of the screened compounds against the tested cell lines was compared to vinblastine sulfate as a reference anticancer drug. The screened compounds showed promising anticancer activities against the tested cell lines, especially PC3 (prostate carcinoma). The most potent anticancer activities against the PC3 cell line were exhibited by hydrazone **3a** (Table 1).

As an extension of the work, all the pyrimidine candidates were evaluated for their cytotoxic effects on the growth of the human prostate carcinoma cell line PC3 and normal human lung fibroblasts (MRC-5) using the MTT assay. The cytotoxicity results of this assay are shown in Table 1. Most of the screened pyrimidines have anti-proliferative activity on the growth of the PC3 cell line. However, the highest antitumor activity was recorded for compounds **3d** and **3b** with an IC50 value of **17** and **21** µM, respectively (~2 to 3 × fold more potent than vinblastine sulfate). Moreover, the introduction of a nitro group at the C-4 position of the benzylidene moiety (**3i**) or the existence of an additional double bond as in compound (**3h**) led to good or moderate cytotoxic activity with IC_50_ values of **37** and **44** µM, respectively. Furthermore, the presence of a cholro group at the C-4 position of the benzylidene moiety intensely dropped the cytotoxic activity, indicating that the position of substituent on the benzylidene and the electronic effect had a remarkable effect on the cytotoxic activity. Alternatively, the replacement of the benzylidene group with the oxoindolylidene group (compound **4**) led to an inactive compound with cytotoxicity up to 603 µM. The results of the cytotoxicity assay in the current study showed that the benzylidene hydrazino moiety is essential for determining the inhibitory potency of the pyrimidine candidates.

For assessing both efficiency and safety of pyrimidine candidates, the selectivity of the tested compounds towards cancer cells relative to normal cells has been measured.

The selectivity index (SI) is a measure of the drug candidate’s selectivity towards cancer cells rather than normal cells. It can be calculated as the ratio between the tested compound CC_50_ on normal cells and its IC_50_ on cancer cells. Generally, a drug with an SI greater than 3 is considered highly selective. Accordingly, the screened compounds **3b**, **3f**, **3g**, **3h**, and **5** were assessed according to the values of their selectivity index (SI) and were found more selective and safer than vinblastine sulfate as illustrated in Table 2.

The morphological changes for the two cell lines (PC3 and MRC) were observed and illustrated in Table 3. Treatment of the cells by the active and most safe compounds **3b**, **3h**, **3g**, **3f**, and **5** revealed that the cancer cells shrank and lost their normal shape relative to untreated cells, indicating initiation of apoptosis of the cancer cell line. On the other hand, there was no significant apoptotic effect on the normal MRC cells.

### 2.3. Physicochemical Properties

The physicochemical properties of all the pyrimidine ligands **1**–**6** were assessed using in silico computational tools [31]. The oral bioavailability of a drug candidate can be predicted via the Lipinski rule of five (RO5). According to the (RO5) rule, good oral bioavailability for a compound must match the following physicochemical parameter values, including: molecular weight (MW) less than or equal to 500 g/mol; a partition coefficient clogP not greater than five, number of hydrogen bond donors (HBD) (NH and OH groups) less than or equal to five, and number of hydrogen bond acceptors (HBA) (O and N atoms) not exceeding ten. The synthesized pyrimidines (**1**–**6**) were validated through Lipinski’s rule of five descriptors. Interestingly, RO5 analysis results (Table 4) revealed that all the ligands, as well as reference drugs, were completely in agreement with Lipinski’s rule of five. Additional valuable assessment for oral bioavailability is the calculation of Veber descriptors, namely: NROTB (number of rotatable bonds) and TPSA (topological polar surface area). Rotatable bonds are defined as any single bond, not in a ring, bound to a nonterminal heavy atom; omitted from the counted C–N amide bonds due to their high rotational energy barrier. The NROTB is an estimate of the molecular flexibility of the compound. Moreover, NROTB should not be greater than ten for good oral bioavailability**.** All pyrimidine ligands **1**–**6** as well as the reference drugs displayed less than ten rotatable bonds. Furthermore, TPSA can be defined as the surface area occupied by nitrogen, oxygen atoms, and the hydrogens bonded to them and it is a significant indicator of the hydrogen bonding capacity and polarity. TPSA is a good descriptor for drug absorption including the blood–brain barrier, intestinal absorption, bioavailability, and penetration. Calculated TPSA values of ligands **1**–**6** along with the reference drugs were smaller than 140 A^2^ which agrees to the known value of the majority of drugs, accordingly, assume promising oral bioavailability. Additionally, the drug-likeness score was similarly calculated as a collective descriptor of physicochemical properties, pharmacokinetics, and pharmacodynamics of the synthesized pyrimidine ligands such as electronic distribution, hydrophobicity, molecule size, flexibility, and hydrogen bonding characteristics. Drug-likeness with positive values is more likely to be drug-like. As shown in Table 3, ligands **3c**, **3f**, **3g**, and **4** exhibited positive scores, indicating a hopeful drug-like behavior.

### 2.4. Molecular Docking

Inhibition of cell apoptosis (programmed cell death) is significant in the progression of prostate cancer. Bcl-2 is one of the most important genes that regulate apoptosis [32,33,34]**.** The current molecular docking study provides structural insights about the Bcl-2 protein and prediction for its molecular interactions with the designed pyrimidine ligands. All designed pyrimidines candidates got docked onto the ligand binding domain of the Bcl-2 protein with a negative dock energy value as shown in Table 5. Molecular interaction studies of the Bcl-2 protein-bound ligands (Figure 4) showed that Bcl-2 protein had SER 60, ARG 129, GLN 118, LEU 59, HIS 120, LYS 58, and GLU 136’ as potential binding sites. The chemical nature of binding site residues suggested that both hydrophobic and polar interactions are essential for proper anchoring of the ligands into the ligand binding pocket. The polar interactions are represented by residues SER 60, GLN 118, LEU 59, and HIS 120, whereas ARG 129, GLU 136, and LYS 58 residues are responsible mainly for hydrophobic interactions. Moreover, the docking results revealed that eleven of the pyrimidine candidates binded to more than one binding site (Figure 4), while three ligands had only one binding site residue each. A well-distinguished binding energy value was found for compound **5** (−5.8815 kcal/mol) which interacted via pi–cation interaction with the active site ARG 129. Moreover, compound **4** exhibited slightly lower binding energy than **5** (binding energy ≈ −5.77). This compound exhibited H-bonding (3.02, 3.25 Å) interaction between the oxygen of its oxoindolylidene moiety and amino acid residues at the backbone of the protein namely, LEU 59 and LYS 58, respectively. Alternatively, compound **6**, which showed the lowest binding energy among the nominated pyrimidines candidates, only displayed hydrogen-bonding interaction via its lactam group with SER 60 active site. It could be concluded that the introduction of the benzylidene moiety improved the inhibitory activity. Moreover, both the electronic factor and position of the substituent of the benzylidene hydrazino group (ligands **3a**–**i**) affect the binding affinity to the target protein. Finally, results of the molecular docking simulation predicted that the newly synthesized pyrimidines would be able to interact and bind with the Bcl-2 with good binding affinity.

## 3. Materials and Methods

### 3.1. Instruments and Apparatus

Melting points were determined by the MEL-TEMP II melting point apparatus in open-glass capillaries and were uncorrected. The IR spectra were recorded as potassium bromide (KBr) discs on a Perkin-Elmer FT-IR (Fourier-transform infrared spectroscopy), Faculty of Science, Alexandria University. The NMR spectra were carried out at ambient temperature (~25 °C) on a (JEOL) 500 MHz spectrophotometer using tetramethylsilane (TMS) as an internal standard, NMR Unit, Faculty of Science, Mansoura University. Chemical shift was recorded as δ values in parts per million (ppm), and the signals were reported as s (singlet), d (doublet), t (triplet), and m (multiplet). Elemental analyses were analyzed at the Micro Analytical Unit, Faculty of Science, Cairo University. The biological evaluation was carried out in the Medical Mycology Laboratory of the Regional Center for Mycology and Biotechnology of Al-Azhar University, Cairo, Egypt.

### 3.2. Molecular Docking

Molecular docking simulations were performed to achieve the mode of interaction of prepared pyrimidines with the binding pocket of the Bcl-2 receptor ligand binding domain. The newly released crystal structure of the Bcl-2 receptor ligand binding domain as a receptor was retrieved from Protein Data Bank (www.rcsb.org, accessed on 25 June 2022) with PDB ID: 5VO4 [35]. Molecular Operating Environment (MOE) (version 2015.10) software was used to prepare the input files and analyze the result. All water molecules, ligands, and ions were removed from the PDB file for the preparation of the protein input file. The active site was selected utilizing the ‘Site Finder’ MOE 2015.10 feature. Prior to docking, the pyrimidines structures were subjected to energy minimization and geometry optimization before docking. Docking simulations were conducted several times with various fitting protocols to observe the best molecular interactions and free binding energies. All docking results were sorted by scoring binding energy.

### 3.3. Cell Lines and Reagents

MRC-5 (normal human Lung fibroblast cells) were obtained from VACSERA Tissue Culture Unit. PC-3 cells (human prostate carcinoma cell line) were obtained from the American Type Culture Collection (ATCC, Rockville, MD, USA). The normal human lung fibroblasts (MRC-5) cells were propagated in Dulbecco’s Modified Eagle’s Medium (DMEM) supplemented with 10% heat-inactivated fetal bovine serum, 1% L-glutamine, HEPES buffer, and 50µg/mL gentamycin. The human prostate carcinoma cell line PC3 cells were grown on RPMI-1640 medium supplemented with 10% inactivated fetal calf serum and 50 µg/mL gentamycin. All cells were maintained at 37 °C in a humidified atmosphere with 5% CO_2_ and were subcultured two times a week. Chemicals used: Dimethyl sulfoxide (DMSO), MTT (3-(4,5-dimethylthiazol-2-yl)-2,5-diphenyl tetrazolium bromide), crystal violet, and trypan blue dye were purchased from Sigma (St. Louis, MO, USA). Fetal bovine serum, DMEM, RPMI-1640, HEPES buffer solution, L-glutamine, gentamycin, and 0.25% trypsin-EDTA were purchased from Lonza. Crystal violet stain (1%): composed of 0.5% (*w*/*v*) crystal violet and 50% methanol then made up to volume with deionized water and filtered through a Whatman No.1 filter paper.

### 3.4. Cytotoxicity Evaluation Using Viability Assay

For antitumor assays, the human prostate carcinoma cell line PC3 was suspended in medium at a concentration of 5 × 10^4^ cell/well in Corning^®^ 96-well tissue culture plates, then incubated for 24 h [36]. The tested compounds were then added into 96-well plates (three replicates) to achieve eight concentrations for each compound. Six vehicle controls with media or 0.5 % DMSO were run for each 96-well plate as a control. After incubating for 24 h, the numbers of viable cells were determined by the MTT test. Briefly, the media was removed from the 96-well plate and replaced with 100 µL of fresh culture RPMI-1640 medium without phenol red then 10 µL of the 12 mM MTT stock solution (5 mg of MTT in 1 mL of PBS) to each well including the untreated controls. The 96-well plates were then incubated at 37 °C and 5% CO_2_ for 4 h. An 85 µL aliquot of the media was removed from the wells, and 50 µL of DMSO was added to each well and mixed thoroughly with a pipette and incubated at 37 °C for 10 min. Then, the optical density was measured at 590 nm with the microplate reader (Sunrise, Tecan Inc, Morrisville, NC, USA) to determine the number of viable cells and the percentage of viability was calculated as [(ODt/ODc)] × 100% where ODt is the mean optical density of wells treated with the tested sample and ODc is the mean optical density of untreated cells. The relation between surviving cells and drug concentration was plotted to obtain the survival curve of each tumor cell line after treatment with the specified compound. The 50% inhibitory concentration (IC_50_), the concentration required to cause toxic effects in 50% of intact cells, was estimated from graphic plots of the dose–response curve for each conc. using GraphPad Prism software (San Diego, CA, USA).

For cytotoxicity assay, the normal human lung fibroblasts (MRC-5) cells were seeded in a 96-well plate at a cell concentration of 1 × 10^4^ cells per well in 100 µL of growth medium. Fresh medium containing different concentrations of the test sample was added after 24 h of seeding. Serial two-fold dilutions of the tested chemical compound were added to confluent cell monolayers dispensed into 96-well, flat-bottomed microtiter plates (Falcon; Franklin, NJ, USA) using a multichannel pipette. The microtiter plates were incubated at 37 °C in a humidified incubator with 5% CO_2_ for a period of 24 h. Three wells were used for each concentration of the test sample. Control cells were incubated without the test sample and with or without DMSO. The little percentage of DMSO present in the wells (maximal 0.1%) was found not to affect the experiment. After incubation of the cells, viable cell yield was determined by a colorimetric method. After incubating for 24 h, the numbers of viable cells were determined by the MTT test. The cytotoxic concentration (CC_50_), the concentration required to cause toxic effects in 50% of intact cells, was estimated from graphic plots of the dose–response curve for each conc. using GraphPad Prism software (San Diego, CA. USA).

### 3.5. Microscopic Observation of the Tumor and Normal Cells Treated with the Purified Compounds

This experiment was performed as previously described in the procedure for antitumor activity. After the end of the treatment at 50 µg/mL concentration, the plates were inverted to remove the medium, the wells were washed three times with 300 µL of phosphate buffered saline (pH 7.2), and then the cells were fixed to the plate with 10% formalin for 15 min at room temperature. The fixed cells were then stained with 100 µL of 0.25% crystal violet for 20 min. The stain was removed, and the plates were rinsed using deionized water to remove the excess stain and then allowed to dry. The cellular morphology was observed using an inverted microscope (CKX41; Olympus, Tokyo, Japan) equipped with a digital microscopy camera to capture the images representing the morphological changes compared to control cells. The cytopathic effects (morphological alterations) were microscopically detected at 100×.

### 3.6. Synthesis of Pyrimidines ***1**–**6***

#### 3.6.1. 2-Thio-6-oxo-4-phenyl-1,6-dihydropyrimidine-5-carbonitrile (**1**)

Ethanolic solution of ethyl cyanoacetate (0.01 mol), thiourea (0.01 mol), benzaldehyde (0.01 mol), and potassium carbonate (0.01 mol) was heated at reflux for 19 h. The precipitated potassium salt was filtered off and washed with ethanol, yellow powder; yield: 98%; m.p.: 385–388 °C. Then, a hot aqueous solution of the potassium salt (at 80 °C) was acidified with acetic acid and stirred for 5 min. The deposited precipitate was collected and washed well with water, dried, and crystallized from ethanol. Yellowish-white crystal; yield: 80%; m.p.: 311–314 °C [lit. 300–302 °C] [37]. ^1^H NMR (500 MHz: DMSO-*d6*): δ 12.82 (brs, 2*H*, 2NH), 7.66 (d, 2*H*, o-Ar-H), 7.63 (t, 1*H*, p-Ar-H), 7.56 ppm (t, 2*H*, m-Ar-H).

#### 3.6.2. 2-Hydrazino-6-oxo-4-phenyl-1,6-dihydropyrimidine-5-carbonitrile (**2**)

Ethanolic solution of compound **1** (0.01 mol) and hydrazine hydrate (0.05 mol) was heated at reflux for 20 h. The precipitate was filtrated and washed with ethanol without the need for further purification. Yellow crystal, yield: 75%; m.p.: 227–230 °C [lit. 229–231 °C] [33]. ^1^H NMR (500 MHz: DMSO-*d6*): δ 12.72 (broad s, 1*H*, NH-pyrimidine), 7.77 (d, 2*H*, phenyl 2*H*), 7.49 (m, 3*H*, phenyl 3*H*), 7.01 ppm (brs, 3*H*, NH-NH2) [37].

#### 3.6.3. 2-(2-Arylidenehydrazinyl)-6-oxo-4-phenyl-1,6-dihydropyrimidine-5-carbonitrile (**3a**–**i**, **4**)

##### 2-(2-Benzylidenehydrazinyl)-6-oxo-4-phenyl-1,6-dihydropyrimidine-5-carbonitrile (**3a**)

Pale yellow crystals, 0.22 g (70%) yield; m.p.: 279–299 °C. IR (KBr): 3233(N-H), 3061(sp2 =C-H), 2204 (CN), 1657(C=O), and 1588 (C=N) cm^−^1. ^1^H NMR (500 MHz: DMSO-*d6*): δ 12.59 (s, 1*H*, NH), 12.45 (s, 1*H*, NH), 8.18 (s, 1*H*, CH=N), 8.03 (dd, *J* = 6.5, 3.0 Hz, 2*H*, Ar-H), 7.85 (d, *J* = 6.8 Hz, 2*H*, Ar-H), 7.60–7.49 (m, 3*H*, Ar-H), 7.46–7.39 (m, 2*H*, Ar-H) ppm. ^13^C-NMR (APT) (125 MHz: DMSO-*d6*): δ 171.65, 162.29, 153.78, 148.24, 136.53, 133.97, 131.96, 131.29, 129.28, 129.02, 128.86, 128.56, 86.95 ppm. C18H13N5O requires: C: 68.55; H: 4.16; N: 22.21% found: C: 68.39; H: 4.33; N: 22.49 %.

##### 2-(2-(4-Bromobenzylidene)hydrazinyl)-6-oxo-4-phenyl-1,6-dihydropyrimidine-5-carbonitri-le (**3b**)

Yellow crystals, 0.28 g (72%) yield; m.p.: 284–285 °C. IR (KBr): 3445 (N-H), 3040 (sp2 =C-H), 2214 (CN), 1652(C=O), 1594 (C=N) cm^−1^. ^1^H NMR (500 MHz: DMSO-*d6*): δ 12.61 (brs, 2*H*, 2NH), 8.14 (s, 1*H*, CH=N), 8.00 (d, *J* = 8.4 Hz, 2*H*, Ar-H), 7.84 (d, *J* = 9.2 Hz, 2*H*, Ar-H), 7.62 (d, *J* = 8.4 Hz, 2*H*, Ar-H), 7.58–7.51 (m, 3*H*, Ar-H) ppm. ^13^C-NMR (APT) (125 MHz: DMSO-*d6*): δ 171.18, 162.26, 154.03, 146.42, 136.64, 133.50, 132.06, 131.71, 130.50, 128.87, 128.83, 124.37, 117.61, 87.28 ppm. C_18_H_12_BrN_5_O requires: C: 54.83; H: 3.07; N: 17.76% found: C: 55.07; H: 3.24; N: 17.53%.

##### 2-(2-(4-Chlorobenzylidene)hydrazinyl)-6-oxo-4-phenyl-1,6-dihydropyrimidine-5-carbonitri-le (**3c**)

Yellow crystals, 0.26 g (75%) yield; m.p.: 280–283 °C. IR (KBr): 3443 (N-H), 3091 (sp2 =C-H), 2213 (CN), 1653 (C=O), 1581 (C=N) cm^−1^. ^1^H NMR (500 MHz: DMSO-*d6*): δ 12.55 (brs, 2*H*, 2NH), 8.15 (s, 1*H*, CH=N), 8.06 (d, *J* = 8.5 Hz, 2*H*, Ar-H), 7.84 (d, *J* = 6.8 Hz, 2*H*, Ar-H), 7.59–7.50 (m, 3*H*, Ar-H), 7.48 (d, *J* = 8.5 Hz, 2*H*, Ar-H) ppm. ^13^C-NMR (APT) (125 MHz: DMSO-*d6*): δ 171.18, 162.28, 154.05, 146.30, 136.65, 135.45, 133.18, 131.70, 130.29, 129.15, 128.86, 128.82, 117.61, 87.25 ppm. C_18_H_12_ClN_5_O requires: C: 61.80; H: 3.46; N: 20.02% found: C: 62.06; H: 3.67; N: 20.27%.

##### 2-(2-(2-Chlorobenzylidene)hydrazinyl)-6-oxo-4-phenyl-1,6-dihydropyrimidine-5-carbonitri-le (**3d**)

Yellow crystals, 0.23 g (67%) yield; m.p.: 305 °C. IR (KBr): 3441 (N-H), 3080 (sp2 =C-H), 2217 (CN), 1667(C=O), 1605 (C=N) cm^−1^. 1*H* NMR (500 MHz: DMSO-*d6*): δ 12.65 (brs, 2*H*, 2NH), 8.69 (d, *J* = 7.4 Hz, 1*H*, Ar-H), 8.60 (s, 1*H*, CH=N), 7.85 (d, *J* = 6.9 Hz, 2*H*, Ar-H), 7.63–7.35 (m, 6*H*, Ar-H) ppm. ^13^C-NMR (APT) (125 MHz: DMSO-*d6*) δ 171.14, 162.24, 153.99, 143.44, 136.60, 133.76, 132.40, 131.75, 131.35, 130.19, 129.11, 128.89, 128.84, 127.80, 117.55, 87.57 ppm. C_18_H_12_ClN_5_O requires: C: 61.80; H: 3.46; N: 20.02% found: C: 62.07; H: 3.69; N: 20.18%.

##### 2-(2-(2-Methylbenzylidene)hydrazinyl)-6-oxo-4-phenyl-1,6-dihydropyrimidine-5-carboni-trile (**3e**)

Yellow crystals, 0.22 g (69%) yield; m.p.: 285–288 °C. IR (KBr): 3215 (N-H), 3058 (sp^2^ =C-H), 2922 (sp^3^-C-H), 2212 (CN), 1662 (C=O), 1589 (C=N) cm^−1^. ^1^H NMR (500 MHz: DMSO-*d6*): δ 12.43 (s, 2*H*, 2NH), 8.52 (s, 1*H*, CH=N), 8.46 (d, *J* = 7.7 Hz, 1*H*, Ar-H), 7.85 (d, *J* = 9.3 Hz, 2*H*, Ar-H), 7.64–7.44 (m, 3*H*, Ar-H), 7.32 (t, *J* = 7.7 Hz, 1*H*, Ar-H), 7.22–7.26 (m, 2*H*, Ar-H), 2.39 (s, 3*H*,CH_3_) ppm. ^13^C-NMR (APT) (125 MHz: DMSO-*d6* δ 171.16, 162.23, 153.92, 146.02, 137.57, 136.71, 132.05, 131.69, 131.09, 130.81, 128.87, 128.82, 127.32, 126.50, 117.67, 86.98, 19.10 ppm. C_19_H_15_N_5_O requires: C: 69.27; H: 4.59; N: 21.26% found: C: 69.06; H: 4.80; N: 21.38%.

##### 2-(2-(3-Methoxybenzylidene)hydrazinyl)-6-oxo-4-phenyl-1,6-dihydropyrimidine-5-carbo-nitrile (**3f**)

Yellow crystals, 0.23 g (68%) yield; m.p.: 340 °C. IR (KBr): 3443 (N-H), 3170 (sp^2^ =C-H), 2957 (sp^3^-C-H), 2212 (CN), 1662 (C=O), 1594 (C=N) cm^−1^. ^1^H NMR (500 MHz: DMSO-*d6*): δ 12.59 (brs, 1*H*, NH), 12.39 (brs, 1*H*, NH), 8.15 (s, 1*H*, CH=N), 7.85 (dd, *J* = 7.9, 1.3 Hz, 2*H*, Ar-H), 7.68 (s, 1*H*, Ar-H), 7.61–7.50 (m, 3*H*, Ar-H), 7.47 (d, *J* = 7.6 Hz, 1*H*, Ar-H), 7.34 (t, *J* = 7.2 Hz, 1*H*, Ar-H), 7.01 (dd, *J* = 8.1, 2.3 Hz, 1*H*, Ar-H), 3.82 (s, 3*H*, O-CH_3_) ppm. ^13^C-NMR (APT) (125 MHz: DMSO-*d6* δ 171.19, 162.25, 160.09, 154.09, 147.69, 136.67, 135.50, 131.69, 130.19, 128.86, 128.83, 121.77, 117.66, 117.07, 112.75, 87.10, 55.93 ppm. C_19_H_15_N_5_O_2_ requires: C: 66.06; H: 4.38; N: 20.28% found: C: 65.88; H: 4.65; N: 20.49%

##### 2-(2-(4-Hydroxybenzylidene)hydrazinyl)-6-oxo-4-phenyl-1,6-dihydropyrimidine-5-carboni-trile (**3g**)

Orange crystals, 0.24 g (73%) yield; m.p.: 291–294 °C. IR (KBr): 3439 (OH), 3352 (N-H**),** 3020 (sp^2^ =C-H), 2237 (CN), 1695 (C=O), 1611 (C=N) cm^−1^. ^1^H NMR (500 MHz: DMSO-*d6*): δ 12.42 (brs, 1*H*, NH), 12.24 (brs, 1*H*, NH), 9.99 (s, 1*H*, OH), 8.07 (s, 1*H*, CH=N), 7.86–7.82 (m, 4*H*, Ar-H), 7.61–7.44 (m, 3*H*, Ar-H), 6.80 (d, *J* = 8.7 Hz, 2*H*, Ar-H) ppm. ^13^C-NMR (APT) (125 MHz: DMSO-*d6*): δ 171.20, 162.27, 160.35, 153.86, 147.98, 136.80, 131.61, 130.56, 128.83, 128.80, 125.25, 117.80, 115.96, 86.39 ppm. C_18_H_13_N_5_O_2_ requires: C: 65.24; H: 3.96; N: 21.14% found: C: 65.49; H: 3.74; N: 21.37%.

##### 6-Oxo-4-phenyl-2-(2-(3-phenylallylidene)hydrazinyl)-1,6-dihydropyrimidine-5-carbonitrile (**3h**)

Orange crystals, 0.24 g (71%) yield; m.p.: 264–267 °C. IR (KBr): 3207 (N-H), 3030 (sp^2^ =C-H), 2211 (CN), 1655 (C=O), 1626 (C=C), 1575 (C=N) cm^−1^. ^1^H NMR (500 MHz: DMSO-*d6*): δ 12.53 (brs, 1*H*, NH), 11.85 (brs, 1*H*, NH), 8.00 (d, *J* = 9.2 Hz, 1*H*, C-CH=N), 7.83 (d, *J* = 8.4 Hz, 2*H*, Ar-H), 7.60–7.50 (m, 5*H*, Ar-H), 7.41 (t, *J* = 7.5 Hz, 2*H*, Ar-H), 7.34 (t, *J* = 7.3 Hz, 1*H*, Ar-H), 7.15 (d, *J* = 16.1 Hz, 1*H*, Ph-CH=CH), 6.98 (dd, *J* = 16.2, 9.2 Hz, 1*H*, CH=CH-CH) ppm. ^13^C-NMR (APT) (125 MHz: DMSO-*d6* δ 170.67, 161.31, 153.30, 148.76, 139.93, 136.17, 135.70, 131.17, 129.19, 128.99, 128.34, 128.30, 127.10, 124.63, 117.05, 86.56 ppm. C_20_H_15_N_5_O requires: C: 70.35; H: 4.43; N: 20.51% found: C: 70.18; H: 4.69; N: 20.41%.

##### 2-(2-(4-Nitrobenzylidene)hydrazinyl)-6-oxo-4-phenyl-1,6-dihydropyrimidine-5-carbonitrile (**3i**)

Red needles, 0.28 g (97.2%) yield; m.p.: 292–295 °C. IR (KBr): 3311 (N-H), 3102 (sp^2^ =C-H), 2216 (CN), 1656 (C=O), 1582 (C=N) and 1502, 1339 (NO_2_, asymmetric and symmetric stretching) cm^−1^. ^1^H NMR (500 MHz: DMSO-*d6*): δ 12.80 (brs, 2*H*, 2NH), 8.33 (d, *J* = 8.0 Hz, 2*H*, Ar-H), 8.25–8.26 (m, 3*H*, Ar-H), 7.86 (d, *J* = 7.2 Hz, 2*H*, Ar-H), 7.53–7.58 (m, 3*H*, Ar-H) ppm. ^13^C-NMR (APT) (125 MHz: DMSO-*d6*): δ 170.61, 161.72, 153.53, 147.96, 144.54, 139.98, 136.02, 131.25, 128.99, 128.36, 128.32, 123.67, 116.93, 87.46 ppm. C_19_H_12_N_8_O_3_ requires: C: 56.99; H: 3.02; N: 27.99% found: C: 57.16; H: 3.26; N: 28.61%.

##### Synthesis of 6-Oxo-2-(2-(isatin-3-ylidene)hydrazinyl)-4-phenyl-1,6-dihydropyrimidine-5-carbonitrile (**4**)

Yellow crystals, 0.23 g (65%) yield; m.p.: 361 °C. IR (KBr): 3439 (N-H), 3095 (sp^2^ =C-H), 2225 (CN), 1685 (C=O), 1627 (C=O), 1594 (C=N) cm^-1^. ^1^H NMR (500 MHz: DMSO-*d6*): δ 12.78 (brs, 2*H*, 2NH-pyrimidine-NH and hydrazino-H), 11.31 (s, 1*H*, NH-isatin), 7.90 (d, *J* = 7.1 Hz, 2*H*, Ph-H), 7.80 (d, *J* = 7.4 Hz, 1*H*, isatin-CH), 7.64–7.51 (m, 3*H*, Ph-H), 7.38 (t, *J* = 7.5 Hz, 1*H*, isatin-CH), 7.12 (t, *J* = 7.6 Hz, 1*H*, isatin-CH), 6.93 (d, *J* = 7.9 Hz, 1*H*, isatin-CH) ppm. ^13^C-NMR (APT) (125 MHz: DMSO-*d6* δ 170.19, 162.84, 153.18, 142.61, 136.54, 135.55, 132.02, 131.60, 128.56, 128.49, 122.65, 122.00, 119.65, 116.50, 111.22, 89.92 ppm. C_19_H_12_N_6_O_2_ requires: C: 64.03; H: 3.40; N: 23.58% found: C: 64.21; H: 3.68; N: 23.75%.

#### 3.6.4. 6-Oxo-3,4,8-triphenyl-4,6-dihydro-1*H*-pyrimido[2,1-c]-1,2,4-triazine-7-carbonitrile (**5**)

A mixture of compound **2** (0.227 g, 0.001 mol), benzoin (0.21 g, 0.001 mol), and pyridine:acetic anhydride (20 mL) (1:1) was heated at reflux for 5 h. The reaction mixture was allowed to cool, poured onto ice water, and neutralized with dilute HCl. The solid product was filtered off and recrystallized from ethanol as yellowish yellow crystals; 0.22 g (55%) yield; m.p.: 285–290 °C. IR (KBr): 3351(N-H), 3038 (sp^2^ =C-H), 2921 (sp^3^-C-H), 2216 (CN), 1600 (C=O), and 1588 (C=N) cm^−1^. ^1^H NMR (500 MHz: DMSO-*d6*): δ 12.45 (s, 1*H*, NH), 7.83 (d, *J* = 7.3 Hz, 2*H*, Ar-H), 7.57–5.07 (m, 4*H*, Ar-H), 7.42–7.31 (m, 7*H*, Ar-H), 7.31–7.20 (m, 2*H*, Ar-H), 6.46 (s, 1*H*, CH) ppm. ^13^C-NMR (APT) (125 MHz: DMSO-*d6* δ 161.89, 159.32, 141.41, 139.60, 135.48, 131.87, 130.39, 130.10, 129.32, 129.17, 128.95, 128.81, 128.56, 128.31, 128.11, 127.57, 126.93, 117.39, 76.60 ppm. C_25_H_17_N_5_O requires: C: 74.41; H: 4.25; N: 17.36% found: C: 74.68; H: 4.39; N: 17.60%.

#### 3.6.5. Synthesis of 3-Hydroxy-6-oxo-8-phenyl-2,6-dihydro-1*H*-pyrimido[2,1-c]-1,2,4-triazine-7-carbonitrile (**6**)

A solution of compound **2** (0.227 g, 0.001 mol), chloroacetyl chloride (0.08 mL, 0.001 mol) in DMF (20 mL), was heated at reflux for 4 h after cooling the reaction mixture was poured onto ice. The solid obtained was filtered and crystallized from methanol to give yellowish green powder, 0.16 g (65%) yield; m.p.: 215–222 °C. IR (KBr): 3416 (OH), 3300 (N-H), 3148 (sp^2^ =C-H), 2207 (CN),1649 (C=O) and 1595 (C=N) cm^−1^. ^1^H NMR (500 MHz: DMSO-*d6*): δ 11.00 (brs, 1*H*, NH), 7.78 (brs, 1*H*, OH), 7.66 (d, *J* = 6.9 Hz, 2*H*, Ar-H), 7.58–7.46 (m, 4*H*, Ar-H and triazine-CH) ppm. ^13^C-NMR (APT) (125 MHz: DMSO-*d6*): δ 169.18, 164.41, 156.02, 136.30, 130.02, 128.02, 127.94, 127.81, 118.43 ppm. C_13_H_9_N_5_O_2_ requires: C: 58.41; H: 3.40; N: 26.21% found: C: 58.90; H: 3.54; N: 25.75%.

## 4. Conclusions

Preliminary in vitro cytotoxicity investigation for the newly synthesized pyrimidines using MTT assay revealed promising activity against human prostate carcinoma cell line PC3. Compound **3b** was found to be two-fold more potent than the reference vinblastine sulfate with an IC50 value of 21 µM. Moreover, the selectivity index (SI) showed that **3b** is 7 times more selective towards cancer cells rather than normal cells. Thus, it is safer than the reference vinblastine sulfate. Physicochemical properties of the pyrimidine-based compounds showed that the compounds fall in the criteria of RO5, thus having the potential to be orally bioavailable. Molecular docking simulation predicted that the newly synthesized pyrimidines would be able to interact and bind with the Bcl-2 with good binding affinity, which could explain a possible mechanism of action via Bcl-2, which is not confirmed.

## Data Availability

The data presented in this study are available on request from the corresponding author.

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
