# Peer review of "Pyrimidines-Based Heterocyclic Compounds: Synthesis, Cytoxicity Evaluation and Molecular Docking"

_molecules, 2022, doi:10.3390/molecules27154912_

Round 1

Reviewer 1 Report

A point-by-point answer to reviewers' comments is not provided. However, it seems that only few of these comments have been addressed. 

The authors modulated their discussion in order to make clearer that the activity of compounds on Bcl-2 is only predicted. 

Still, english language should be carefully checked, as some repetitions, mistakes or wrong punctuation are present.

In Scheme 1, details about the reagents and conditions are still missing in the caption (such as reaction time, yields, etc).

Units for IC50 values are often missing and where present are reported as uM/mL (better use mM or uM).

Refluxed was not changed as suggested. It has to be done since this word doesn't exist.

No suggested reference was inserted.

The changes only slightly improved the paper. Additional work is required before the manuscript can be considered acceptable for publication.

Author Response

Dear Reviewer,

I would like first to thank the Referee for his valuable and accurate comments that helped us to revise our manuscript more thoroughly.  All his suggestions have been considered in the revised manuscript via "Track Changes" function in the Microsoft Word.

  • The authors modulated their discussion in order to make clearer that the activity of compounds on Bcl-2 is only predicted. 

Thanks a lot. That was according to the reviewer’s valuable suggestion

  • Still, English language should be carefully checked, as some repetitions, mistakes or wrong punctuation are present.

The English was double checked.

  • In Scheme 1, details about the reagents and conditions are still missing in the caption (such as reaction time, yields, etc).

Has been addressed

  • Units for IC50 values are often missing and where present are reported as uM/mL (better use mM or uM).

Has been addressed, units for IC50 values are reported as uM.

  • Refluxed was not changed as suggested. It has to be done since this word doesn't exist.

Has been addressed, refluxed has been changed by heated at reflux.

  • No suggested reference was inserted.

All the suggested ref. has been inserted

Reviewer 2 Report

The authors made some change to their manuscript, but the reviewer didn’t find much improvement. The author failed to address the major issues the reviewer raised in their original manuscript. The authors need to revise the manuscript based on the reviewer’s comments instead of just putting the reviewer’s comments into their manuscript. Therefore, the reviewer still cannot support its publication in Molecules. The reviewer suggests the authors submit their manuscript to journals such as journal of heterocyclic chemistry, or journal of enzyme inhibition and medicinal chemistry etc.

In addition to the issues the reviewer’s previous comments, the originality/novelty of the manuscript is low. Compound 2 was found in 9 publications; 3a, 3c, and 3i can be found in the following two papers, https://doi.org/10.1515/znb-1998-1017, https://doi.org/10.1002/jhet.5570220141, and the rest compounds 3 are just simple close analogues; 3g is a close to the compounds in this paper (https://doi.org/10.1002/ardp.201700403); 4 is close to compounds in this paper (https://doi.org/10.1016/j.bioorg.2020.104051).

I don’t think the above-mentioned papers are cited in the manuscript.

Line 480, “In addition, all the designed pyrimidine candidates exhibited good binding affinity with the anticancer target protein receptor” unless the authors can use biochemical assays to support their claim, such statements are not acceptable.

The IC50 value unit is µM instead of µM/mL. µM means µmol/L; µg/mL is still used in SI, whichever is used, it should be consistent.

Author Response

Dear Reviewer,

I would like first to thank the Referee for his valuable and accurate comments that helped us to revise our manuscript more thoroughly.  All his suggestions have been considered in the revised manuscript via "Track Changes" function in the Microsoft Word.

  • In addition to the issues the reviewer’s previous comments, the originality/novelty of the manuscript is low. Compound 2was found in 9 publications; 3a3c, and 3i can be found in the following two papers, https://doi.org/10.1515/znb-1998-1017, https://doi.org/10.1002/jhet.5570220141, and the rest compounds 3 are just simple close analogues; 3g is a close to the compounds in this paper (https://doi.org/10.1002/ardp.201700403); 4 is close to compounds in this paper (https://doi.org/10.1016/j.bioorg.2020.104051).

Regarding the novelty of the work,

The current work includes

  • Synthesis of number of pyrimidine-based compounds, as mentioned above some of these compounds have been reported before (2, 3a, 3c and 3i). However, all the other compounds are new, and all the compounds have been fully characterized using IR, H-NMR, C-NMR and elemental analysis.

Compound 2 is just starting for the synthesis of all other compounds. And it has been referenced in the text.

Even (compounds 3a, 3c and 3i) were previously reported but the current work provides their full characterization which was missing in the old references.

Compounds 3g or may be close in their structure to other compounds but they are still new and never been prepared before.

Moreover, the authors in the current work didn’t report a new synthetic methodology, but we aimed to develop new pyrimidine derivatives, that are easily synthesized from readily accessible starting materials at mild conditions with good yields.

  • In vitro cytotoxicity investigation for the synthesized pyrimidines (2-6), which was never been reported before for the studied compounds
  • Molecular docking simulation to predict the ability of the pyrimidines under investigation to interact and bind with the Bcl-2.
  • Assessment for, the physicochemical properties of all pyrimidine ligands using in silico computational tools.

Although the current  work doesn’t provide a new methodology, however the new synthesized pyrimidine and the new observation for their cytotoxicity, molecular docking and physicochemical properties  may contribute to scientific progress.

  • Line 480, “In addition, all the designed pyrimidine candidates exhibited good binding affinity with the anticancer target protein receptor” unless the authors can use biochemical assays to support their claim, such statements are not acceptable.

The authors modulated their discussion in order to make clearer that the activity of compounds on Bcl-2 is only predicted. 

And this sentence in the both discussion and conclusion has been changed to ((Molecular docking simulation predicted  that the newly synthesized pyrimidines would be able to interact and bind with the Bcl-2 with good binding affinity))

  • The IC50value unit is µM instead of µM/mL. µM means µmol/L; µg/mL is still used in SI, whichever is used, it should be consistent.

Has been addressed, units for IC50 values are corrected and reported as uM.

  • The authors made some change to their manuscript, but the reviewer didn’t find much improvement. The author failed to address the major issues the reviewer raised in their original manuscript. The authors need to revise the manuscript based on the reviewer’s comments instead of just putting the reviewer’s comments into their manuscript. Therefore, the reviewer still cannot support its publication in Molecules. The reviewer suggests the authors submit their manuscript to journals such as journal of heterocyclic chemistry, or journal of enzyme inhibition and medicinal chemistry

Herein, you can find a point-by-point answer to reviewers' comments to follow up the changes done in the manuscript

  1. Pyrimidines are a critical pharmacophore existing in numerous drugs. The authors use erlotinib as an example. Erlotinib is an EGFR inhibitor used for treatment of lung and pancreatic cancers.  There is no relevance between erlotinib and the work presented in this manuscript, you cannot randomly pick a drug consisting of a pyrimidine as an example; alternatively, you can show more drugs containing a pyrimidine used for different indications.

Has been addressed in both introduction and in figure 1.

  1. The design of the compounds lacks logic. You cannot say because most anticancer drugs have more than one aromatic ring, so you want to include more rings. It all depends on the target. The ring introduced should enhance protein binding, improve compound’s physicochemical properties, etc

Has been addressed, the authors no more claim the anticancer activity  of the prepared compounds. Instead preliminary study for their cytotoxicity has been discussed, accordingly, the title and the discussion  of the manuscript has been changed. and the discussion related to the design also has been changed in the text.

  1. Bcl2 doesn’t seem to be a well validated therapeutic target in PC3 cells. Actually, bcl2 expression in PC3 cells is very low. In addition, I cannot accept the authors’ claim that the compounds synthesized are bcl2 inhibitors. The authors use docking to show that the compounds can be docked to bcl2 protein. However, docking results can only serve as complementary data to support/explain observed experimental results. For example, if you already have experimental data showing the compounds inhibit bcl2 activity or can bind with bcl2, then you use docking to understand how the compounds bind to bcl2. Docking data alone can be extremely misleading, you can dock a lot of compounds into a proposed protein pocket and get “good” binding results. Unless the authors can provide biochemical assay data that the compounds can bind to bcl2, otherwise the claim is not acceptable.

Has been addressed. The authors modulated their discussion in order to make clearer that the activity of compounds on Bcl-2 is only predicted. 

  1. It’s barely seen in papers people use “ug/ml” to describe the compounds inhibition activity. When you compare the activities, you would rather compare by molecule-to molecule instead of mass-to-mass because their molecular weights are different. Also, as observed in Table 1, the compounds are apparently more active in HepG-2 cells. I don’t understand why the authors choose PC3 for further study. Probably because the compounds are too less active than vinblastine sulfate. Moreover, vinblastine sulfate is a chemotherapy drug, which apparently has a low selectivity index; so it doesn’t make the compounds look safer even if they have a higher SI than vinblastine sulfate, however the majority of the compounds’ SI are quite low as well.

Has been addressed. All the “ug/ml has been converted into µM“

  1. There is a misinterpretation of Lipinski’s rule of five. The rule of five is just an observation that most orally administered drugs fall in these criteria. It doesn’t mean the compounds “must” follow these rules to be orally bioavailable; and, vice versa, compounds possessing these properties are also not granted to be promising drug candidates. You can only say the synthesized compounds fall in these criteria of RO5, thus having the potential to be orally bioavailable.

Has been addressed. It has been  reported in the abstract that the prepared compounds fall in these criteria of RO5, thus having the potential to be orally bioavailable

Minor issues:

  1. The chemistry section is too long. The reactions have already been widely reported, I didn’t see much novelty in the synthesis, so it’s unnecessary to spend a whole page to report the compounds’ characterizing data. It’s not attractive to readers as well.

Chemistry part is essential to be clarified, probably it would be significant for readers  interested in organic synthesis or  heterocyclic chemistry.

  1. IC50values need to be reported in molar concentration (e.g. uM).

Done

  1. Data in Figure 3 were not discussed in the manuscript. 

Done

Round 2

Reviewer 2 Report

compounds number in Table 1 and 5 should be bolded.

inconsistency of text fonts needs to be corrected.

This manuscript is a resubmission of an earlier submission. The following is a list of the peer review reports and author responses from that submission.

Round 1

Reviewer 1 Report

In the manuscript entitled 'Design, Synthesis, Molecular docking and Evaluation of anticancer activity of new pyrimidines-based compounds' authors describe synthesis of several pyrimidines-based compounds and test their cytostatic effects on prostate cancer (PC3) cells and lung fibroblasts. They find that some of these compounds inhibit the growth of cancer line. Besides this, they use computer modelling to model the interaction of these compounds with anti-apoptotic protein Bcl-2. Indeed, if these compounds, would turn out to be real inhibitors of Bcl-2, this result would be of great interest.

As far as I can judge, the experiments are conducted and interpreted correctly. The language of the manuscript is good, however, some minor editing (mostly punctuation and capitalization) will be required.

Minor concerns:

- it has not been established that tested compounds in fact bind to Bcl-2. I suggest to slightly change the language to reflect this.

- line 203: 'While three had only one active site... should rather be 'While three had only one binding site..'

Reviewer 2 Report

The manuscript presented by El-Atawy, etc described the synthesis of a series of dihydropyrimidinones and their anticancer activities in a few human cancer cell lines. However, the overall quality of the manuscript is too low to be published in Molecules.

Critical scientific flaws as below:

1. Pyrimidines are a critical pharmacophore existing in numerous drugs. The authors use erlotinib as an example. Erlotinib is an EGFR inhibitor used for treatment of lung and pancreatic cancers.  There is no relevance between erlotinib and the work presented in this manuscript, you cannot randomly pick a drug consisting of a pyrimidine as an example; alternatively, you can show more drugs containing a pyrimidine used for different indications.

2. The design of the compounds lacks logic. You cannot say because most anticancer drugs have more than one aromatic ring, so you want to include more rings. It all depends on the target. The ring introduced should enhance protein binding, improve compound’s physicochemical properties, etc

3. Bcl2 doesn’t seem to be a well validated therapeutic target in PC3 cells. Actually, bcl2 expression in PC3 cells is very low. In addition, I cannot accept the authors’ claim that the compounds synthesized are bcl2 inhibitors. The authors use docking to show that the compounds can be docked to bcl2 protein. However, docking results can only serve as complementary data to support/explain observed experimental results. For example, if you already have experimental data showing the compounds inhibit bcl2 activity or can bind with bcl2, then you use docking to understand how the compounds bind to bcl2. Docking data alone can be extremely misleading, you can dock a lot of compounds into a proposed protein pocket and get “good” binding results. Unless the authors can provide biochemical assay data that the compounds can bind to bcl2, otherwise the claim is not acceptable.

4. It’s barely seen in papers people use “ug/ml” to describe the compounds inhibition activity. When you compare the activities, you would rather compare by molecule-to molecule instead of mass-to-mass because their molecular weights are different. Also, as observed in Table 1, the compounds are apparently more active in HepG-2 cells. I don’t understand why the authors choose PC3 for further study. Probably because the compounds are too less active than vinblastine sulfate. Moreover, vinblastine sulfate is a chemotherapy drug, which apparently has a low selectivity index; so it doesn’t make the compounds look safer even if they have a higher SI than vinblastine sulfate, however the majority of the compounds’ SI are quite low as well.

5. There is a misinterpretation of Lipinski’s rule of five. The rule of five is just an observation that most orally administered drugs fall in these criteria. It doesn’t mean the compounds “must” follow these rules to be orally bioavailable; and, vice versa, compounds possessing these properties are also not granted to be promising drug candidates. You can only say the synthesized compounds fall in these criteria of RO5, thus having the potential to be orally bioavailable.

Minor issues:

1. The chemistry section is too long. The reactions have already been widely reported, I didn’t see much novelty in the synthesis, so it’s unnecessary to spend a whole page to report the compounds’ characterizing data. It’s not attractive to readers as well.

2. IC50 values need to be reported in molar concentration (e.g. uM).

3. Data in Figure 3 were not discussed in the manuscript. 

Reviewer 3 Report

In their paper, the authors have explored the cytotoxic activity of some pyrimidines-based compounds. Design, synthesis, MTT assay, and in silico prediction of the possible interaction with Bcl-2 have been reported.

The manuscript is nicely set up, but several points need to be addressed.

In the abstract, it is not clear if compounds have been synthesized according to the molecular docking analysis or because of their structural relation to the pyrimidine scaffold.

The rationale behind compounds 3, 5 and 6 is reported but no space has been given to compound 4. It is not even in figure 2. Moreover, here the structures of compounds 5 and 6 are wrong (it seems that a double bond is missing).

In the chemistry scheme, reagents and conditions should be provided in the caption. Remove them from the arrows, and list them below, including all the reagents, solvents, temperature, time of reaction and yields. The scheme is also a bit stretched. Please make sure is in acs1996 style.

While discussing cytotoxicity and anticancer activity, IC50s should have the unit after. I would also express the data as mM, µM or nM.

Although computational analysis has been performed, it is not possible to certainly assume that these compounds bind to Bcl-2. If the authors want to state that, they should perform direct binding assays on the protein. Otherwise, they should rephrase and give less emphasis to the in-silico results, just pointing out a possible mechanism of action via Bcl-2, which is not confirmed. In addition, the authors discuss about the apoptosis linked to Bcl-2 interaction, but their compounds are only screened for the cytotoxic activity in the MTT assay, no annexin staining has been performed therefore there is no solid proof of a pro-apoptotic effect.

While discussing about Bcl-2, references of its role in apoptosis (such as Nature Reviews Molecular Cell Biology, 2019, 20, 175–193) and a few papers dealing with literature compounds previously reported (Front. Pharmacol. 2019, 10:391 and European Journal of Medicinal Chemistry, 2016, 117, 301-320) should be inserted.

When describing a procedure at high T°, the term “refluxed” is not correct. It should be “heated at reflux”.

The solvent(s) for the crystallization procedure should be specified (in the general procedure or for each compound if different).

Along the text, some little mistakes (such as double spacing, repetition, etc.) are highlighted in red (see attached pdf file).
